# Gold Nanoparticles: Biosynthesis and Potential of Biomedical Application

**DOI:** 10.3390/jfb12040070

**Published:** 2021-12-03

**Authors:** Ekaterina O. Mikhailova

**Affiliations:** Institute of Innovation Management, Kazan National Research Technological University, K. Marx Street 68, 420015 Kazan, Russia; katyushka.glukhova@gmail.com

**Keywords:** gold nanoparticles, green synthesis, capping agents, antimicrobial activity, anticancer activity, antiviral activity

## Abstract

Gold nanoparticles (AuNPs) are extremely promising objects for solving a wide range of biomedical problems. The gold nanoparticles production by biological method (“green synthesis”) is eco-friendly and allows minimization of the amount of harmful chemical and toxic byproducts. This review is devoted to the AuNPs biosynthesis peculiarities using various living organisms (bacteria, fungi, algae, and plants). The participation of various biomolecules in the AuNPs synthesis and the influence of size, shapes, and capping agents on the functionalities are described. The proposed action mechanisms on target cells are highlighted. The biological activities of “green” AuNPs (antimicrobial, anticancer, antiviral, etc.) and the possibilities of their further biomedical application are also discussed.

## 1. Introduction

The production of the substances using biological synthesis is of particular interest to receiving new therapeutic compounds and environment safety. The last decades were marked by a huge number of works devoted to nanoparticles produced by so-called “green synthesis”. The metal-based nanoparticles are silver, gold, platinum, nickel, manganese, titanium, and zinc nanoparticles [1]. Biological properties with a “+” sign against pathogenic microorganisms, cancer cells, various protozoa, helminths, etc., are presented [2,3,4,5,6]. Despite the fact that silver nanoparticles occupy the lion’s share of this topic’s research [7], other nanoparticles also seem to be interesting objects. Gold nanoparticles are undoubtedly the second most popular nanoparticles due to their distinctive physicochemical properties [8,9]. Gold has been used for both therapeutic and aesthetic purposes since ancient times. The term “soluble gold” appeared in China and Egypt in the 4th or 5th century BC [10]. The most famous example is the Lycurgus Cup, which shows a different color depending on the dichroic effect achieved by making the glass with proportions of gold and silver nanoparticles dispersed in the colloidal form [11]. In the Middle Ages, gold popularity was explained by magical and healing properties such as treatment of heart and infectious diseases, cancers, and a beneficial organism effect [12]. The gold antibacterial properties were first described by the outstanding microbiologist Robert Koch in 1890, who studied the low concentrations effect of potassium cyanide on tuberculosis bacilli [13]. The first report about gold nanoparticles—AuNPs— was published by Faraday in 1857, who studied gold nanoparticles in a colloidal (dispersed) system and described in detail their optical features, such as the light-scattering properties of suspended gold microparticles [14]. Then the theory for scattering and absorption by spherical particles was formulated by G. Mie [15]. At the end of the XIX–beginning of the XX century, R.A. Zsigmondy was the first to describe the methods of colloidal gold synthesis with different particle sizes [16]. The rapid growth of nanotechnology in the late 20th and early 21st centuries has made AuNPs one of the most intensively studied objects to solve fundamental and applied problems in medicine. Physical and chemical methods were used to obtain gold nanoparticles, but such synthesis of NPs was accompanied by using highly toxic chemicals and the formation of dangerous byproducts [17]. Unlike chemical and physical technics “green” synthesis did not have these disadvantages. The term “green” synthesis has several meanings: environmentally friendly, economical, high yield, single-step option, and safe for humans. Different methods are applied to characterize biogenic AuNPs. The shape and size of synthesized “green” AuNPs are defined by Scanning Electron Microscopy (SEM) and Transmission electron microscopy (TEM) [18,19]. The proposal coating biomolecules attached to the AuNPs surface, responsible for its reduction and stabilization, are identified by FTIR [20,21]. The wide possibilities of AuNPs use were discovered due to their diverse properties. Antimicrobial, antiviral, and anticancer activities were found to open a new chapter in the treatment of various diseases [22,23,24]. The property of surface plasmon resonance (SPR) gives a possibility to use gold nanoparticles as sensors in biological and chemical sciences [25,26]. Important AuNPs’ aspects are their geometric shape, size, as well as the parameters of biosynthesis (temperature, pH), and the biological object used as a “green factory” [27]. The last is one of the most significant, because diversified cellular compounds—proteins, enzymes, acids, etc., can play an important role in the characteristic features of AuNPs received from specific objects. For medical use, this means an absence of toxic effect on healthy human cells, as well as the purposefulness of the effect of nanoparticles. This review is devoted to the amazing properties of gold nanoparticles, but also to their synthesis mechanisms and interaction with living objects, which can become the basis of potential multipolar application.

## 2. Properties of Gold Nanoparticles

### 2.1. AuNPs Biosynthesis

Chemical and physical methods were traditionally applied for the synthesis of gold nanoparticles. However, their use is accompanied by several drawbacks. For the chemical methods, the main disadvantages (for example, using Citrate reduction, 1-amino-2-naphthol-4-sulfonic acid (ANSA) [28,29]) are supposed to use highly toxic reagents, environmental pollution, carcinogenic solvents, contamination of precursor. On the other hand, physical methods (for example, a laser irradiation method) require expensive equipment and high energy consumption [30]. In addition, the low stability of AuNPs, difficulties in controlling crystal growth, and particle aggregation make the above methods less advantageous. Currently, the “green” method of AuNPs synthesis is attracting more attention due to the expansion of nanotechnology capabilities. The use of non-toxic agents without additional stabilizers and reducing agents, renewable materials, low energy expenditure, and ecological safety are the key factors of biological synthesis popularity. The living organisms’ great diversity allows the production of specific, practice-oriented gold nanoparticles. Moreover, the biomolecules involved in the biosynthesis by bacteria, fungi, algae, and plants have a positive effect both on the synthesis process and on the resulting AuNPs.

*The biosynthesis mechanism*. The production of gold nanoparticles is a sufficiently simple process that does not require an increase in temperature and pressure. The general scheme assumes the following: the biological extract (bacterial, fungal, plant, etc.) is added dropwise to the HAuCI_4_ salt solution and mixed well to initiate the AuNPs synthesis process [31]. The color change of the resulting solution indicates the nanoparticles production. Notwithstanding that many publications are illustrating the AuNPs synthesis using different organisms (bacteria, algae, fungi, plants), the mechanism of the biogenic process is not fully understood. The biosynthesis takes place in two steps: at the first, Au^3+^ is reduced to Au^0^, and at the second, agglomeration and stabilization result in the AuNPs formation (Figure 1) [32]. Interestingly, a wide variety of bio-compounds (enzymes, phenols, sugars, etc.) can participate both in the gold reduction and in the stabilization and capping of nanoparticles [33,34,35].

*Biosynthesis by bacteria*. Microorganisms can act as a potential “factory” for gold nanoparticles production [36]. The biosynthesis mechanism was found to be both extracellular and intracellular for bacteria according to the location of AuNPs production (Figure 2) [37,38].

However, the extracellular synthesis of gold nanoparticles is the most common [38]. Gold ions are first trapped on the surface or inside the microbial cells and then reduced to nanoparticles in the presence of enzymes [36]. It is supposed that the enzymatic way is one of the best possible routes for AuNPs synthesis [36]. The enzyme nitrate reductase was shown to play a vital role in the gold ions reduction [39,40]. For example, the AuNPs biosynthesis in the bacteria *Stenotrophomonas maltophilia*, *Rhodopseudomonas capsulate*, luminescent bacteria *Pseudomonas putida*, and *Pseudomonas fluorescence* is associated with the enzyme NADPH-dependent reductase, which converts Au^3+^ to Au^0^ through the enzymatic process of metal reduction using electron transfer [41,42,43,44]. The extracellular proteolytical nature is hypothesized for the AuNPs biosynthesis process in *Actinobacter* spp. [45]. The presence of AuNPs on the membrane inner side suggests that some gold ions (Au^3+^) can cross the cell barrier through the ion transfer channel and are reduced by enzymes on the cytoplasmic membrane and inside the cytoplasm [41]. The positively charged metal ions transport, with the help of negatively charged proteins or microbial enzymes binding to them on the cell wall surface or in the cytoplasm, subsequently forming AuNPs of various sizes and shapes, is an intracellular mechanism of AuNPs biosynthesis [39,46,47]. In addition, this process can be mediated through ion pumps, carrier-mediated transport, endocytosis, ion channels, or lipid permeation [48]. Thus, AuNPs synthesis by non-pathogenic bacteria *Deinococcus radiodurans*, known for their resistance to radiation and oxidants, showed that the presence of a wide range of antioxidants (for example, carotenoid, pyrroloquinoline-quinone, and phosphoproteins) for protecting against oxidative damage of nucleic acids and proteins, can provide a microenvironment to facilitate the reduction of Au (III) and the AuNPs formation [49]. Gold nanoparticles were distributed throughout the cell wall, cytosol, and extracellular space. The intracellular AuNPs presence suggests that gold ions can be transported into cells and converted into AuNPs [49]. Another example of an intracellular synthesis mechanism is biosynthesis by *Lactobacillus kimchius* [50]. In addition, it is supposed that NADH-dependent enzymes and sugars secreted by microorganisms on the cell surface are responsible for the Au^3+^ reduction, while proteins and amino acid residues inside cells can be stabilizing agents for nanoparticles [50,51].

*Biosynthesis by fungi*. The fungal synthesis of gold nanoparticles can also be both extracellular and intracellular. The intracellular mechanism can be realized by reducing sugars, proteins such as ATPase, glyceraldehyde-3-phosphate dehydrogenase, and 3-glucan-binding proteins involved in the energy metabolism of fungal cells [40,52]. Au^3+^ diffuse through the cell membrane and are reduced by systolic redox mediators. However, it is unclear whether the diffusion of the Au^3+^ ions occurs through the membrane by active bioaccumulation or passive biosorption [53,54]. Interestingly, the fungal ultrathin slices research indicated the AuNPs concentration in the vacuoles of cells [40]. Extracellular formation of gold nanoparticles occurs by adsorption of AuCl^4−^ ions on cell wall enzymes by electrostatic interaction with positively charged groups [55]. Regardless, NADPH-dependent oxidoreductases either on the cell surface or in the cytoplasm are the key enzymes in AuNPs biosynthesis, apparently, as in the case of other nanoparticles (for example, AgNPs) [56,57,58,59]. Das et al. determined that NADH acted as a cofactor of a protein and/or an enzyme (for example, glutathione reductase) responsible for the gold ions reduction [60,61]. A glutathione-like compound, phytochelatin of *Candida albicans,* was shown as another alternative compound directly involved in the AuNPs synthesis [62]. In the presence of glutathione, Au ions initiate the synthesis of phytochelatin, then Au^3+^ ions are reduced to AuNPs [62]. Another remarkable example of the AuNPs synthesis may be biosynthesis due to phenol oxidases–Mn peroxidases, laccases, and tyrosinases in xylotrophic basidiomycetes both intracellularly and extracellularly [63]. Notably, the melanin was found to be involved in the biosynthesis of gold nanoparticles by *Yarrowia lipolytica* [64].

*Biosynthesis by algae*. Another original object for production and studying various properties of AuNPs are algae. Being a source of specific compounds typical only for this group of organisms (for example, fucoidan, neutral glucan, and guluronic and mannuronic acid residues containing alginic acid) with a wide range of biological activities (antibacterial, anticoagulant, and antifouling activity), algae have enormous biomedical significance. The synthesis process can proceed by extracellular and intracellular mechanisms [65]. Thus, sulfonated polysaccharide compounds and amide bond protein molecules can be involved in the reduction of gold ions to nanoparticles and AuNPs stabilization in an aqueous medium using *Turbinaria conoides* [66]. In addition, a synthesis mechanism implying electrostatic interactions between gold anions and functional groups of algae was proposed [67]. [AuCl_4_^−^] bound to positively charged functional groups, such as amino groups (-NH_2_), on the algae surface, and after 40 and 50 min, algae extracts reduced Au(III) to gold nanoparticles [67]. Proteins and polysaccharides (alginate and sulfated fucoidans) in the cellular biomass of brown algae provide many binding sites for heavy metals due to the presence of hydroxyl groups [67,68]. At the initial stage, stoichiometric interaction between cell components and metal ions was observed, followed by the accumulation of heavy metals at the binding sites [67,68]. Chakraborty et al. suggest that secreted algal enzymes take part in the AuNPs biosynthesis [69,70], and one of the crucial roles seems to be played by NADPH-dependent reductase, which can act as NADH electron carrier and can efficiently convert Au ions into AuNPs through an enzymatically mediated electron transfer process occurring in the inner membrane matrix of mitochondria [70].

*Biosynthesis by plants*. A lot of plant species growing on our planet are an inexhaustible resource of helpful substances used in medical practice since ancient times. Therefore, plants are the most popular “biofactories” for AuNPs. Interestingly, a wide variety of biomolecules can participate in the gold nanoparticles biosynthesis, and the process of synthesis and AuNPs formation, apparently, is dependent on the nature of using plant extract. Thus, phenolic acids in the extract may be responsible for the reduction in metal ions and corresponding nanoparticles formation [71]. Flavonoids can be of importance in the biogenic synthesis (Au^3+^ can form an intermediate complex with a free radical of flavonoids, which subsequently undergoes oxidation to keto-forms, followed by reduction of trivalent gold to AuNPs) [72,73,74,75], other phenolic compounds (for example, salicin may be responsible for the AuNPs formation through hydroxyl group and glucoside bonds, promoting the reduction of Au^3+^ to Au^0^ and AuNPs stabilization) [75], terpenoids (may play a role in the metal ions reduction by oxidation of aldehyde groups in molecules to carboxylic acids) [76], and polyphenols [77,78]. Thus, the deprotonation of the hydroxyl groups in the polyphenolic molecules was demonstrated for gold nanoparticles synthesis using *Mimosa tenuiflora* extract, i.e., the first stage of the reduction process leads to the transfer of electrons from the deprotonated hydroxyl group to Au^3+^ ions [79]. Au^3+^ ions are reduced to Au^0^ metal atoms, and the polyphenolic ring is oxidized [79]. The possibility of tannins [80], alkaloids [81] and polyols [82] involvement was revealed for AuNPs biosynthesis. The hydroxyl groups in polyols were found to be oxidized to carboxylate groups during the reduction of Au^3+^ to Au^0^ [82]. Metal ions reduction and formation of corresponding nanoparticles may be associated with plant extracts sugars [83]. The reduction site of the polysaccharide can give away amino groups that might increase the stability of metal NPs. Thus, both amino group and carbohydrates firmly bind to the hydrophilic surface provided by AuNPs. Hydroxyl groups of polysaccharides are oxidized to carbonyl groups, thereby reducing Au from Au (III) to Au (0) [84]. Additionally, proteins with a high molecular weight can be attributed to important molecules related to the biosynthesis of gold nanoparticles [85,86]. Gold reduction and stabilization of synthesized gold nanoparticles by some exotic biomolecules, for example, citrulline from watermelon rind was discovered [87].

### 2.2. The AuNPs Morphology (Shape and Size)

The nature of the AuNPs absorption spectrum is known to be largely dependent on the size and shape [88,89]. Gold nanoparticles are widely applied due to their electrical and optical properties, and the ability to form strong complexes with biomolecules [90]. The particle size and the rate of AuNPs formation can be manipulated by controlling parameters such as pH, temperature, and gold concentration [91]. Table 1, Table 2, Table 3 and Table 4 provide some information about “green” AuNPs synthesized by microorganisms, fungi, algae, and plants [41,42,43,49,71,72,73,74,75,78,79,80,81,82,87,92,93,94,95,96,97,98,99,100,101,102,103,104,105,106,107,108,109,110,111,112,113,114,115,116,117,118,119]. AuNPs are very diverse in shape, although spherical nanoparticles are considered the dominant variant. Depending on the production method, gold nanoparticles can take different forms: triangle, hexagon, octahedron, cells, nanospheres, wells, stars, and nanorods [85]. The shape of nanoparticles is important because it greatly affects their physical properties. According to the Mi theory, the frequency of the plasmon band varies from spherical to non-spherical nanoparticles of various shapes (rods, prisms, triangles, cubes, shells) [120]. In addition, dependence between the extract concentration and the predominant form of gold nanoparticles was found: at lower extract concentration more triangular and prismatic nanoparticles are synthesized than hexagonal and spherical ones [121]. A decrease in the reaction time leads to obtaining a larger number of hexagonal and triangular AuNPs [80,122]. The sizes also differ in fairly large limits. For example, 10 nm spherical AuNPs have surface plasmonic absorption at around 520 nm [123]. An increase in the particle size results in a deflection in the absorption spectrum—the maximum absorption for 48.3 and 99.4 nm AuNPs is in the range of 533 and 575 nm, respectively. Changing the shape of nanostructures on nanorods can shift absorption to the near-infrared region of the spectrum [123]. In addition, small nanoparticles easily attach to the cell, and antibacterial activity grows up [109]. The biosynthesis and formation of gold nanoparticles are influenced by temperature and pH solution [85].

### 2.3. Capping and Stabilizing Agents

The next stages of biogenic synthesis are the AuNPs capping and stabilization. The gold nanoparticles obtained by a non-biological method can interact with biological fluids and come into contact with tissues exposed to active biomolecules that surround them and form a “crown” (“corona” in Latin). Thus, the nanoparticles acquired a biological component: the so-called protein corona (PC) [124,125,126]. Such PC consists mainly of proteins, but the presence of other biomolecules (e.g., sugars, lipids) is also expected [127,128].

A wide variety of compounds play an important role in biogenic gold nanoparticles synthesis. Substances of potential practical significance in combination with gold nanoparticles make such structures useful from many points of view. Moreover, capping and stabilizing agents are extremely important for declining their toxicity, increasing biocompatibility and bioavailability in living cells, as well as practical approaches (antimicrobial activity, anticancer activity, etc.) [33]. Au^0^ has a natural tendency to coagulate, but the molecules from biological (bacterial or fungal cultural medium, plant extract) extracts can cap and stabilize them [129]. Different studies discovered the high values of zeta potential mean that AuNPs are very stable due to the presence of high surface charge preventing agglomeration [130,131]. Biologically synthesized gold nanoparticles can include functional (aromatic, amide, alcohol, etc.) groups playing an important role in AuNPs capping and stabilization [129]. These molecules can enhance the affiliation possibility and action of AuNPs on the bacterial cells [129]. Apparently capping agents have a possibility for selective binding to different types of facets on a nanocrystal to change their specific surface free energies and in their area proportions [131]. It is supposed that the presence of carboxyl or hydroxyls groups in addition to the aromatic rings in different structural units can contribute to the stability of the AuNPs [132]. The three types of nanoparticle stabilization using various capping agents can be highlighted: electrostatic, steric, and unification of steric and electrostatic stabilization [133]. However, the compounds for the nanoparticles stabilization and final capping are different and specific for “bio-factory”, and especially important in the further practical application of AuNPs [84]. In addition, capping agents can frequently have their biological activities, that can increase the AuNP’s activity. The compounds involved in the capping and stabilization of AuNPs were illustrated in Figure 3.

Protein packaging, as well as specific compounds produced by specific bacteria, are typical for “bacterial” AuNPs. For example, antioxidant compounds–carotenoid, pyrroloquinoline–quinone, and phosphoproteins rich in hydroxyl, phospho-, and amino groups, and also a unique PprI protein implicated in the regulation of the cellular antioxidant system and stress response, were found for *D. radiodurans* (Figure 3a) [49].

According to fungal synthesis, AuNPs can interact with proteins via free amino groups or cysteine residues by electrostatic attraction of negatively charged carboxyl or carbonyl groups, forming a coating on nanoparticles to prevent agglomeration, stabilizing AuNPs [134]. These results suggest that hydroxyl, amine, and carboxyl groups play an important part in the stabilization of synthesized AuNPs. The presence of amide bonds for keeping amino acid residues of proteins, such as tryptophan/tyrosine, secreted extracellularly, can stabilize fungal-mediated AuNPs [97]. Proteins attachment on the surface of AuNPs can also be implemented using van der Waals forces [96]. Besides, phosphate bonds, polypeptides [135], primary, secondary, and tertiary amides [136], aromatic and aliphatic amines [97], polysaccharides, and lipids [137] are supposed to participate in gold nanoparticles capping (Figure 3b). Interestingly, it seems that AuNPs capping is produced exclusively by large biomolecules (more than 3 kDa) [96].

Proteins participate in the capping and stabilization of algal AuNPs [66]. The carbonyl group of amino acids having a strong ability to bind metal can be used as capping compounds [66]. Sulfonated polysaccharide compounds [66], hydroxyl functional group in alcohol and phenolic compounds, functional group of primary amines [67], different carotenoids (e.g., fucoxanthin) [138], and polysaccharides [70] are found to be capping agents (Figure 3c). More specific compounds, for example, andrographolide, alloaromadendrene oxide, glutamic acid, hexadecanoic acid, oleic acid, 11-eicosenoic acid, stearic acid, gallic acid, epigallocatechin catechin, and epicatechin gallate are determined as capping agents in *G. elongata* [103].

Perhaps, plants are the most exciting objects in capping phytocompounds research. The substances diversity synthesized by various parts of plants, and their participation in AuNPs capping and stabilization, cause particular interest in plant synthesis. FTIR spectroscopy displayed that capping agents cover gold nanoparticles with a thin layer [111]. The presence of hydroxyl and carboxyl ions in biomolecules can lead to the protective layer’s formation on the AuNPs surface stabilizing gold nanoparticles. For example, flavonoids or terpenoids can be adsorbed on the metal nanoparticles’ surface, possibly by interaction through carbonyl groups or π electrons in the absence of other strong ligating agents in sufficient concentration [83,111]. Phenolic compounds, including tannins [79,139,140]; proteins [141], metabolites-having alcohols, aldehydes, ketones [118,142], carbohydrates and saponins [143], alkaloids [144], and fatty acids [108] were discovered to be capping agents (Figure 3d). The detection of other capping bio-compounds is very likely. However, the mechanism of AuNPs biosynthesis was not fully understood, and the involved phytochemicals variety complicates its study. Numerous studies in this area suggest great gold nanoparticles’ potential as safe, non-toxic, and relatively easily received by different kinds of “biofactories”, making a positive contribution to the AuNPs formation (especially capping agents).

### 2.4. Mechanism of AuNPs Action on Cells

Unfortunately, the accurate mechanism of the AuNPs’ effect on the cell is unknown. Nevertheless, a significant data amount was already accumulated, allowing certain conclusions to be made in this area.

*Toxicity for bacterial cells*. The main leitmotivs are the following: on the one hand, gold nanoparticles can attach by adhesion on the cell wall surface and penetrate through the bacterial cell membrane, which leads to integrity and stability disruption of the cytoplasmic membrane, subsequently resulting in bacteria death [44,145]; on the other hand, action is possible and/or through interaction with various cellular organelles and DNA [146]. For example, visible cell surface damage, loss of flagella, cell wall loosening, cytoplasm shrinkage, and release of cellular material were found for AuNPs-treated bacteria *E. coli*, *P. aeruginosa*, *S. aureus* [147]. The shape-dependent antibacterial activity of gold nanoparticles was proposed [147]. The high AuNPs antimicrobial activity is possibly due to their shape, components attached to the surface, and surface charge [147,148]. The damage possibilities of AuNPs are mainly owing to the physical mutilation of bacterial cells, as showed and reinforced by the microscopic observation and nucleic acid leakage [147]. Cell wall damage is the result of electrostatic interaction between positively charged nanoparticles and a negatively charged cell wall. Nanoparticles attached to the cytoderm can penetrate the cell, releasing a large number of ions causing toxicity [31]. Another factor is associated with reactive oxygen species (ROS) [31,149]. The damage results due to the affinity of binding between AuNPs, thiol, and amine groups are what causes the interaction with biomolecules leading to the formation of free radicals [149,150,151]. The release of free radicals was strongly correlated with an increase in membrane permeability and induced various pathogens’ death [152]. The generation of various ROS-O_2_, H_2_O_2_, HO_2_, and OH, causes oxidative stress, leading to lipid peroxidation in the cytoplasmic membrane. Thus, ROS react with macromolecules such as phospholipids, enzymes, and nucleic acids of cytomembranes to form lipid peroxidation products. These increase the cytomembrane permeability leading to structural changes and functions of cells [66,153]. Besides, the lysis of *Str. pneumoniae* was shown according to gold nanoparticles’ interaction with proteins and carbohydrates, resulting in the formation of pores and subsequent cell damage [154]. The ROS formation increased oxidative stress in microbial cells and the release of the intracellular enzyme lactate dehydrogenase into the extracellular medium in vacuoles form [155].

Antibacterial inhibition by biosynthesized AuNPs presumably begins from the binding of extract polyphenols with a microbe’s protein [74]. It changes the membrane potential then reduces the synthase activity of adenosine triphosphate [74]. Cui et al. demonstrated that AuNPs in *E. coli* induce the membrane potential collapse and inhibit the activity of ATPase or, using another pathway, inhibit the binding of tRNA to the ribosome subunit [156]. In addition, AuNPs penetrate through the cell wall into the cell, where they can react with thiol groups to form Au–thiol groups, and thiol groups of cysteine (due to disulfide bridges) can trigger protein folding [153]. The combination of gold and cysteine ions also disrupts microbial respiration and electron transfer systems [31]. In this case, respiratory electron transport is disconnected from oxidative phosphorylation, inhibiting respiratory chain enzymes or breaking membrane permeability for protons and phosphates [103].

AuNPs can bind with bacterial DNA and inhibit the DNA transcription process leading to cell death [157,158]. Furthermore, the free radicals can bind with DNA by interacting with the sulfur and DNA phosphorus group, causing mutations, additions, deletions, single breaks, double-strand breaks, and cross-linking with proteins [81,159]. An interesting mechanism was proposed by Lee et al.: extensive damage of *E. coli* DNA was a result of AuNPs exposure via an apoptosis-like pathway [160]. The programmed prokaryotic cell death was observed in bacteria: cell filamentation caused by cell division stopping during the repair of damaged DNA; the cell membrane was depolarized and DNA fragmented [160]. AuNPs caused cell elongation due to nuclei condensation and fragmentation, signing late apoptosis in *E. coli* [160]. AuNPs induce overexpression of RecA protein and activation of bacterial caspase-like protein(s) in *E. coli* [160]. Thus, gold nanoparticles initiate induction of membrane depolarization, DNA fragmentation, and caspase activation processes similar to apoptotic death in bacteria [160]. Another significant AuNPs effect is the depolarization of bacterial cells associated with Ca^2+^ [161]. The calcium gradient is rigorously supported by channels and transporters system. Depolarization of the plasma membrane potential creates an imbalance between the influx of Ca^2+^ into plasma and the export of Ca^2+^ and leads to a steady increase in the Ca^2+^ level in the cytosol [161,162]. Thus, whereas bacterial differentiation, chemotaxis, pathogenicity, and sporulation are the correlated concentrations of calcium in the cytoplasm, such a process can be an extremely interesting approach in the fight against pathogenic bacteria. Most probably, the presented variants of antibacterial effects can work both together and separately. Additionally, diversified compounds from “bio-factories” can bind to the AuNP’s surface as capping agents and provide high antimicrobial activity [163]. Additionally, shape and size of AuNPs can play an important role in this process [109].

*Toxicity for human cells*. The toxicity analysis of gold nanoparticles is mandatory before using for all kinds of pathologies treatment in humans. AuNPs generally have low acute toxicity both in vitro and in vivo [164,165]. Thus, smaller nanoparticles have greater toxicity [166]. The cytotoxicity of AuNPs is considered to be shape dependent. The spherical AuNPs discovered by Tarantola et al., as a rule, are more toxic and more efficiently absorbed by the cell than rod-shaped particles [167]. At the same time, in [168] nanospheres and nanorods were more toxic than the star, flower-, and prism-shaped AuNPs. AuNPs stars are the most cytotoxic against human cells in [169]. In addition, biomolecules localized on the gold nanoparticles’ surface can also influence these nanomaterials’ toxicity [170,171,172]. AuNP concentration is also important; for example, gold nanoparticles in low concentrations do not exhibit cytotoxic effects in healthy and cancer cell lines [173,174,175,176]. However, due to the different experimental methods’ in vitro models, shapes, sizes, capping agents’ variety, gold nanoparticles functionality, and the variability of cell lines, opinions about AuNPs toxicity can significantly differ [177]. Cell viability and cytotoxicity were evaluated in human umbilical vein endothelial cells (HUVEC), and a moderate cytotoxic effect at 24 and 48 h was found in [79]. However, toxicity does not behave in a dose-dependent manner [79]. On another side, plant-mediated AuNPs were not detected in the nucleus, indicating a small genotoxic potential of nanoparticles or their absence [79]. The toxic effect in vitro was presented for gold nanoparticles synthesized by *B. cereus* and *F. oxysporum* [86]. The low doses of AuNPs were not toxic to tissues, while higher doses disrupted the functioning of all tested organs (brain, liver, spleen, kidney, heart, and lung of rats) in histopathology analysis [178]. Gold ions had a tendency to bind with thiol groups in the liver, induce reducing reactions, transfer glutathione into the gallbladder bile and reduce the concentration of glutathione [179]. The glutathione reduction is significant for the removal of peroxides. Therefore, AuNPs can be toxic in both human and animal tissues, probably according to this mechanism [179]. *Sphaeranthus indicus*-synthesized AuNPs were non-toxic to non-target *Artemia nauplii* microcrustaceans; moreover, all tested animals showed a 100% survival rate [180]. A low cytotoxic effect on the human lung cancer cell line A549 was demonstrated for gold nanoparticles from *Asp. foetidus* [59]. The absence of any significant toxicity in vitro was recognized for AuNPs biosynthesized by *Pistacia atlantica* extract [181]. Comparative analysis of AuNPs effect in vitro on 293 normal cell lines and U87 GBM cells revealed a cytotoxic effect only on U87 GBM cells that had condensed nuclei with fragmented or marginal chromatin structure [182]. Thus, the observed AuNPs effect on various cell types appears very diverse and needs significant further research. Analyzing toxicity is necessary to consider all key characteristics to determine the best working without causing unfavorable effects.

## 3. Biomedical Application of AuNPs

### 3.1. Antimicrobial Activity

*Antibacterial activity*. The high resistance of pathogenic microorganisms to different, and even the most modern antibiotics is becoming an increasingly serious problem for clinical medicine that could be decided using nanoparticles of various metals, including AuNPs. The antimicrobial activity is dependent on the method of synthesis, size, shape, and concentration of biosynthesized gold nanoparticles [183]. The influence mechanism for the pathogenic bacteria of the genus *Bacillus*, *E. coli*, *Streptococcus*, *Staphylococcus*, etc., is still extremely topical [103,106,184]. In addition, a significant point is the belonging of potentially destroyed bacteria to Gram+ or Gram−, according to their cell walls structural features. Although Gram-positive and Gram-negative cell walls are negatively charged with a high-affinity degree to positively charged AuNPs, having a thinner cell wall, Gram-negative bacteria are more simply exposed to AuNPs, while Gram-positive have rigid peptidoglycan layers on their surface, which prevent the AuNPs entry. For example, the inhibitory effect was shown only for Gram-negative bacteria in *E. coli* and *Enterobacter ludwigii*, *B. subtilis*, and *Enterococcus faecalis* research [39]. More considerable antibacterial effect was shown for bio-produced AuNPs compared with chemically synthesized gold nanoparticles [185]. Such antibacterial activity may be due to the synergistic effect of the plant compounds acting as capping agents [185]. AuNPs are a valuable element against bacterial biofilms. The AuNPs weaken the biofilm formation of *Proteus* sp. by inhibiting the production of virulence factors such as exopolysaccharides and metabolic activity such as surface hydrophobicity playing an important role in bacterium–host cell interactions and biofilm architecture in microbes, respectively [186]. In [187], bacterial surface attachment, flagella loss, biofilm assemblage, and clumping inside biofilm are demonstrated as the antibacterial processes.

*Antifungal activity.* Pathogenic fungi (*C. albicans*, *Aspergillus* spp., *Penicillium* spp., *Trichoderma viridae*, etc.) and their associated diseases represent a serious problem for clinical medicine. The emergence of new antibiotic-resistant strains requires the search for new methods of combating these pathogens. Among such potentially applicable substances, gold nanoparticles are emphasized. AuNPs interact with cell wall macromolecules, damaging them and affecting membrane proteins [188]. The inhibition of cell wall β-glucan synthase leads to changes in the cell wall integrity and further cell damage [188,189]. Besides, antifungal activity of gold nanoparticles is possible by increasing the ROS (for instance, in *C. albicans*) [189]. High antifungal activity was observed against *C. tropicalis*, *C. albicans* [190], *A. flavus* and *A. terreus* [191], *A. fumigatus* [192].

### 3.2. Antiviral Activity

Viral diseases have always posed the greatest of human threats. Notwithstanding that the investigation of these infectious agents is very intensive, we still know very little about combating methods. Moreover, for many known viral diseases, neither drugs nor vaccines were not found. Therefore, the struggle methods search against these extremely dangerous organisms stays a very urgent task requiring a prompt, and sometimes immediate decision. Metal nanoparticles are a very promising trend in fighting against various kinds of viruses. It is supposed that AuNPs can bind to a viral particle, blocking the connection with cellular receptors or viral receptors that inhibit viral cycle onset [193]. Aside from that, nanoparticles adsorbed on the cell surface can significantly change the membrane potential, leading to blocking the viral penetration into the cell [193]. Additionally, the inhibition of virus binding and penetration into the host cell, binding to the plasma membrane, inactivation of viral particles before penetration, and interaction with double-stranded DNA were found to be the antiviral mechanism of AuNPs [193]. For instance, gold nanoparticles are offered as an innovative means to counteract the measles virus (MeV) [194]. The active inhibition evidence of MeV replication in Vero cells by AuNPs obtained from garlic extract (*Allium sativa*) was discovered [194]. The interaction of AuNPs and MeV is probably resulting in the viral receptors blocking, preventing cell adsorption and the viral infection onset in the host cell. This type seems to be an ideal antiviral approach that excludes interaction with the cell. Additionally, having high stability and biocompatibility, AuNPs can easily interact with various biologically active compounds of garlic extract, including organosulfur compounds, saponins, phenolic compounds, and polysaccharides [194]. The active components are garlic organosulfur compounds, such as allicin, and products derived from allicin (diallyl sulfide, diallyl disulfide, diallyl trisulfide, ajoene, allyl-cysteine, and allyl-cysteine sulfoxide), which gives additional positive features against viral infection [195]. El-Sheikh et al. identified that AuNPs inhibited the replication of the Herpes Simplex (HSV-1) virus infection to Vero cells in a dose-dependent manner which reduced 90% CPE of HSV-1 at 31.25 μL [196]. Gold nanoparticles synthesized in *Sargassum wightii* extract prevented HSV-1 and HSV-2 viruses’ infection of Vero cells in a dose-dependent manner; moreover, the toxicity absence in high concentrations makes these AuNPs a potential antiviral agent [197]. However, there are other data regarding the gold nanoparticle’s effect on the vital activity of viruses: AuNPs can penetrate through the cell membrane into cells, and then inhibit viral DNA and RNA replication [193]. For example, AuNPs inhibit post-entry Foot-and-Mouth Disease (FMD) virus replication, accompanied by the onset of intracellular viral RNA synthesis, while at non-cytotoxic concentrations, AuNPs do not exhibit extracellular viricidal activity and inhibition of FMD growth in infection early stages, including attachment and penetration [198]. Thus, the proposed mechanism of antiviral activity based on [193,194,195,196,197,198] was demonstrated in Figure 4. Unfortunately, data on the “green” synthesis of gold nanoparticles with antiviral effects are very poor. Most of the works are devoted to chemically produced functional nanoparticles modified with specific molecules. Such complexes can be the basis for drugs’ targeted delivery to organs and tissues, including antiviral fighting.

### 3.3. Antioxidant Activity

Different pathological conditions, including inflammatory processes, atherosclerosis, aging, cancer, and neurodegenerative diseases are highly dependent on oxidative stress caused by ROS, such as hydroxyl, epoxyl, peroxylnitrile, superoxide, and singlet oxygen. The redundant ROS amount or oxidative stress are influencing the host antioxidant system results in nucleic acid damage and enzyme inactivation [199]. Intracellular antioxidant enzymes and intake of antioxidants may help to maintain an adequate antioxidant status in the body [200]. Antioxidants help to reduce DNA damage, malignant transformation, cell damage, and decrease the risk of various pathologies. Antioxidants can decrease oxidative damage directly via reacting with free radicals or indirectly by inhibiting the activity or expression of free radical-generating enzymes or the activity increase or expression of intracellular antioxidant enzymes [200]. The antioxidant activity mechanism includes the following: the antioxidant molecules may directly react with the reactive radicals and destroy them, while they may become new less active free radicals, longer lived, and less dangerous than those radicals they have neutralized [200]. The search for new, safe compounds preventing oxidative damage is extremely meaningful, because despite the presence of effective endogenous antioxidant mechanisms in the human body, the balance between antioxidant action and free radicals’ production is disrupted because of lifestyle changes, radiation, and pollutants. The antioxidant potential of AuNPs produced by “green” synthesis is promising. The widely used and rapid methods for estimating antioxidant activity are the ABTS (2,2-Azino-bis (3-ethylbenzthiazoline-6-sulfonic acid radical) and DPPH (1,1-diphenyl–2–picrylhydrazyl radical) assays [201]. The free radical scavenging activity in vitro was shown for gold nanoparticles produced using extra virgin olive oil [202], nanoparticles synthesized from leaf extract (decoction) of *Antigonon leptopus* [203], *Nerium oleander* leaves extract [204], Kokum fruit extract [205], Cannonball fruit (*Couroupita guianensis*) extract [206], fruit extract of *Hovenia dulcis* [207], *Aconitum toxicum* rhizomes extracts [208], *Artemisia capillaris*, *Portulaca oleracea*, and *Prunella vulgaris* extracts [209], roots of *Angelica pubescens* [210], *Thyme* extract [211], leaves extract of *Origanum vulgare* [212], *Piper longum* fruit extract [213], marine bacterium *Paracoccus haeundaensis* [214], and others. According to most studies, various biomolecules encrusted on the surface of gold nanoparticles increase antioxidant activity. Especially polyphenols: flavins and flavonoids, as well as tannins, being powerful antioxidants themselves, enhance the effect [72,82,208].

### 3.4. Anticancer Activity

The last hundred years were marked by a huge increase in cancers, considered one of the main reasons for death worldwide. Unfortunately, most of the developed drugs and approaches have many side effects. Therefore, the new drugs with low toxicity and synthesized in a “green” way are very prospective anticancer agents. The antitumor effect of gold nanoparticles in vitro was shown for Hela N (Human cervix carcinoma), Hep G2 (human liver cancer), A549 (human lung carcinoma), MCF-7 (breast adenocarcinoma), HCT-11 (colon carcinoma), PANC-1 (human pancreatic cancer), ovarian adenocarcinoma (Caov-4) in a dose-dependent manner [215,216,217,218,219,220,221]. The provided gold nanoparticles effect depending on the shape, size, and chemical composition of the nanoparticle’s surface was discovered in [106,222,223]. Apparently, smaller gold nanoparticles have more antitumor effect due to the larger surface area of smaller NPs [224]. Undoubtedly, capping agents contribute to the antiproliferative activity of AuNPs, participating in the protein’s modification or cell growth enzymes and independently performing anticancer activity [79,224,225]. In addition, the antitumor activity of medicinal plant extracts is expressed by stopping the cell cycle, cell apoptosis, and induction of antiangiogenesis [226,227]. In this way, the plant-synthesized adsorbed active molecules and their therapeutic activity, as well as biocompatible gold nanoparticles are of great importance in anticancer therapy [225]. Although the mechanism of AuNP’s effect on cancer cells is not completely clear, the centerpieces are (a) ROS generation, (b) Glutathione (GSH) oxidation, (c) cell cycle arrest, and (d) caspases [125,126,127,128,129,130,131,132,133,134,135,136,137,138,139,140,141].

The AuNPs’ cytotoxic effect on cancer cells is primarily due to their easy permeability to cellular barriers and strong affinity for various biological macromolecules. As byproducts of normal cellular metabolism, ROS play an important role in cellular signaling pathways such as cell-to-cell signaling, cellular metabolism, cell proliferation, and cell apoptosis. The imbalance in ROS and antioxidant levels plays a critical role in tumor initiation and progression [228]. Gold nanoparticles can induce cytotoxicity through ROS, generating damage to cellular components through intracellular oxidative stress [229,230]. For example, AuNPs increase the ROS production in HeLa cells and probably lead to apoptotic cell death via the mitochondrial-mediated pathway [229]. Decreased mitochondrial membrane permeability and mitochondrial dysfunction leading to apoptosis were discovered for two human renal carcinoma cell lines [228].

Possessing antioxidant properties, GSH not only protects the cell from toxic free radicals but also generally determines the redox characteristics of the intracellular environment. It was found that ROS generation converts GSH to GSSG (Glutathione disulfide) through the oxidation process [231]. Oxidized glutathione is reduced by the enzyme glutathione reductase induced by oxidative stress. The ratio of reduced and oxidized glutathione forms in the cell is one of the most important parameters showing the oxidative stress level. For instance, low GSH levels were observed in cells influenced by star anise-synthesized AuNP [232]. A decrease in the GSH level corresponds to increased oxidative stress [232]. ROS generation in AuNPs-treated cells was also determined in other publications: increased oxidative stress and lipid peroxidation in MRC-5 (human lung fibroblasts); hydrogen-peroxide induced by GSH depletion is generated in HL7702 cells (human liver cell line) [233,234]. Thus, increasing ROS generation and glutathione oxidation may be the basis of AuNPs’ anticancer activity.

Physicochemical interactions of gold atoms with functional groups of intracellular proteins, as well as with nitrogenous bases and phosphate groups in DNA, are another cytotoxic action of gold nanoparticles [235]. The AuNPs influence various cell lines, for example, U87 (human primary glioblastoma cell line) is revealed in DNA degradation, condensed nuclei with fragmented chromatin structure [236,237]. Moreover, the formation of oligo-nucleosomal DNA fragments or ladder owing to DNA fragmentation is widely discussed as a biochemical marker of late apoptosis [238]. Another aspect is the accumulation of AuNPs-treated cells in the sub-G1 phase or G0/G1 phase of the cell cycle, so cell cycle regulation can play a vital role in the apoptosis induction [239]. Thus, a significant percentage of MCF-7 and MDA-MB-231 cells treated by “green” AuNPs were in the G0/G1 and S phases, which may indicate AuNP’s efficiency in inducing cell arrest at various phases of the cell cycle [237,240]. The launch of the apoptosis process–programmed cell death is one of the most important mechanisms of the antitumor effect. It is characterized by morphological changes: cell shrinkage, nuclei fragmentation, and extensive blebbing of the plasma membrane, eventually resulting in apoptotic cells formation that will subsequently be phagocytosed by macrophages [241]. Bcl-2 protein plays an essential role in the apoptosis process, which activates caspase-9 and caspase-3, triggering the apoptosis cascade (with the participation of another caspases-7,8) [242]. Besides, downregulation of p53 (protein p53) may also be a key element of anticancer activity, because it is a transcription factor regulating cell cycle and acting as a suppressor of malignant tumors formation [243]. AuNPs were demonstrated to induce the expression of both p53 and p21 in a concentration-dependent manner in MCF-7 [237]. Thereby, gold nanoparticles are capable of activating cell death through a caspase-mediated apoptotic pathway [244,245,246,247]. Nevertheless, there are still many questions about the anticancer activity of AuNPs; in addition, most studies were made in vitro and need further testing in vivo.

### 3.5. Other Activities

It should be noted that gold nanoparticles have other very useful properties.

*Anti-inflammatory activity*. One of the interesting AuNPs areas is using for anti-inflammatory activity. As mentioned earlier, ROS plays an important role in the activation of many inflammatory mechanisms. That is why gold nanoparticles inhibiting active oxidants are extremely promising in this field. Macrophages play an essential role in the development of inflammatory processes such as phagocytes [248]. LPS-induced RAW 264.7 macrophages are widely used as an in vitro model of inflammation [249]. Thus, the AuNPs influence the expression of iNOS (Inducible nitric oxide synthase) and COX-2 (cyclooxygenase-2) protein in LPS-induced (lipopolysaccharides-induced) RAW 264.7 cells for *Acanthopanacis* cortex extract was determined [250]. AuNPs produced using *Panax ginseng* fresh leaf extract exerted anti-inflammatory effects in LPS-induced RAW 264.7 macrophages by blocking NF-kB signaling (abnormal regulation of NF-kB activity can result in different diseases including inflammatory, cancer, metabolic, and cardiovascular illness) [251].

*Antidiabetic activity*. Despite the World Health Organization regularly developing norms and standards for diabetes diagnosis, treatment, monitoring, and its risk factors, the number of diagnosed cases is constantly increasing from year to year. The conducted experiments demonstrated AuNPs’ possibility to have an antidiabetic effect. Thus, oral AuNPs injection to diabetic animals regulates the metabolic process and restores cholesterol and triglycerides levels to almost normal levels [252]. Rats treated with gold nanoparticles were able to improve body weight by increasing insulin secretion and glycemic control, as well as due to their natural growth [253]. The glucose concentration in the blood serum decreased, favorable changes in body weight occurred, transaminase activity and lipid profile improved in streptozotocin-induced diabetic rats using gold nanoparticles synthesized by *Cassia fistula stem* bark extract [253]. In vitro results showed that AuNPs not only improved insulin secretion induced by di-(2-ethylhexyl) phthalate (DEHP) (DEHP played as a diabetogenic agent by increasing free radicals and decreasing insulin levels finally resulting in loss of pancreatic cells mass) but also protected RIN-5F cells (a clone derived from the RIN-m rat islet line) from toxicity caused by DEHP by increasing cell viability and insulin secretion. AuNPs also prevent oxidative cells damage and normalize the regulation of Bcl-2 (Bcl-2 is a regulatory protein, is involved in apoptotic regulation) family proteins through an unregulated insulin signaling pathway [254,255]. In addition, the antidiabetic activity of AuNPs from *Fritillaria cirrohosa* was shown in preclinical models [256]. Gold nanoparticles from *Ziziphus jujuba* can diminish diabetes complications by lipid peroxidation and oxidative stress decline [257]. Using gold nanoparticles can become the basis for diabetic nephropathy treatment [258]. All these data characterize AuNPs as excellent hypoglycemic agents in diabetes mellitus therapy and related complications.

*Leishmanicidal activity*. The main vectors of Dengue fever and malaria–*Aedes aegypti* and *Anopheles stephensi* mosquitoes represent a very significant threat to the tropical and subtropical population. Gold nanoparticles can help in solving this problem as well. The larvicidal activity was shown for AuNPs from *Jasminum nervosum* leaf extract against filarial and arbovirus vector *Culex quinquefasciatus* [259], against larvae and pupae of the malaria vector *A. stephensi* and the dengue vector *A. aegypti* [260].

*Photothermal therapy*. Photothermal therapy is a minimally invasive technique, which uses hyperthermia generated by photothermal agents from laser energy to kill cancer cells [261]. Hyperthermia was known as one of the most effective radiosensitizers [262]. The nanotechnological idea is to deliver AuNPs specifically to a tumor, apply NIR (near-infrared spectroscopy) light that will predominantly heat only the tumor, and then deliver radiotherapy [263]. Potential gold nanoparticle hyperthermia approaches in cancer treatment may have various advantages [263]: (a) they can be activated via near-infrared (NIR) laser light, creating the ability to penetrate deep into biological tissues; (b) a radiotherapy and hyperthermia combination can lead to higher effectiveness than the use of radiotherapy alone; (c) they can reduce the radiotherapy dose and make it more tumor-specific; (d) direct infusion can reduce common toxicity effect; (e) they can be modified to create multidimensional cancer photothermal therapy and drug delivery systems [264,265]. AuNPs-mediated photothermal therapy combined with checkpoint immunotherapy was discovered to reverse tumor-mediated immunosuppression, thereby leading to the treatment of primary tumors [266]. Green-synthesized curcumin-coated gold nanoparticles can induce apoptotic cell death in photothermal therapy and radiofrequency electric field hyperthermia [267]. Unfortunately, the data about biosynthesized gold nanoparticles and their application in this matter are practically absent.

*Drug delivery*. Gold nanoparticles can be used as a delivery method for various therapeutic agents. Molecules with different functional groups can bind with high affinity on the surface of AuNPs. Capping agents surrounding the AuNPs can be displaced by other functioning thiols or adsorbed ligands through a ligand exchange reaction [38]. AuNPs can bind with other materials covalently and non-covalently [38]. Covalent conjugation stabilizes the conjugates for imaging. Electrostatic interactions, hydrophobic interactions, and specific binding affinity can act as non-covalently binding with AuNPs [38]. Gold nanoparticles can be functionalized by different compounds carrying the healing effect. Coating molecules (for instance, PEG and BSA) are attached to provide a binding surface for specific cells, minimizing, in that way, non-specific targeting on other tissues [268]. For example, PEGylation of gold nanoparticles can minimize macrophages and monocytes uptake, providing them with a cover and prolonging their availability and concentration in tumor tissue [269]. Not only small molecular drugs but also large biomolecules (such as DNA, RNA, peptides, and proteins) are delivered by AuNPs [268]. Anticancer drugs such as doxorubicin, 5-Fluorouracil may be target compounds in delivery by AuNPs [86,237,270,271]. Biosynthesized AuNPs are also used as drug delivery system for cancer therapy in a mouse model [272]. AuNPs modified with tryptophan and 5-aminopurine have excellent antibacterial activity against multidrug-resistant bacteria [273]. Green gold nanoparticles are particularly interesting because, having their capping agents with useful properties, they can be equipped with additional molecules to achieve and increase the therapeutic effect.

*Bio-sensing and Detection*. According to their properties, AuNPs can be used in biosensing. Perfect sensitivity in determining cancerous cells, biological molecules, blood glucose levels, bacteria, viruses, toxins, and pollutants is proved by gold nanoparticles [274]. The optical and electronic properties of AuNPs are used in various cell imaging techniques, such as computed tomography, dark-field microscopy, optical coherence tomography, and Raman spectroscopy. AuNPs properties such as colorimetric, surface plasmon resonance, electrical, electrochemical, and fluorescence can be the base for different kinds of sensors [275]. AuNPs play a crucial role in the analysis called “bio-barcode assay” [276]. This analysis is an ultrasensitive method for detecting target proteins and nucleic acids. The bio-barcode assays are generally based on AuNPs functionalization with many strands of oligonucleotides strands (“barcodes”) and a corresponding recognition agent which can be antibody in terms of protein detection, and a small segment of the barcoded strand in case of nucleic acids detection [276]. Gold nanoparticles are often used as amplifiers in SPR sensors. An important advantage of metal nanoparticles is the dual mechanism of SPR enhancing [277,278,279]. Enhancing of the PPR sensor signal was proposed by Kao et al. in the determination of antibodies against glutamic acid decarboxylase—GAD (glutamic acid decarboxylase—GAD), a marker for the diagnosis of insulin-dependent diabetes [280]. This approach allows decreasing the detection limit of antibodies by four orders of magnitude [280]. The enhanced fluorescent properties of AuNPs have made the detection of aflatoxins easier [281]. AuNPs are of great interest in the colorimetric detection of viruses [282]. The approach is based on the two main techniques: (1) a color amplification technique in which AuNPs are applied to act as direct coloring labels with their characteristic, intense red color; (2) a color changes technique in which a color change from red to purple occurs in response to particle aggregation [283,284]. Gold nanoparticles are applied in microorganisms detection [268]. AuNPs functionalized by oligonucleotides complementary to the unique sequences of the heat-shock protein 70 (HSP 70) of *Cryptosporidium parvum* was used to detect the oocytes of *Cryptosporidium* in a colorimetric assay [285]. *Staphylococcus* enterotoxin B was detected by gold nanoparticle-based chemiluminescence assay [286].

## 4. Conclusions

The eco-friendly mechanism and low toxicity of the applied and obtained compounds have already made the “green” synthesis method so popular. The ease of controlling the size and shape of nanoparticles due to changes in reaction parameters, relatively high reaction speed, and economic efficiency make biosynthesized particles a potential helper in solving a wide range of biomedical tasks. There are some limitations and disadvantages of biological synthesis. Thus, bacterial synthesis requires a long time (ranging from hours to days), delicate preparation stages are necessary to obtain filtrates of mycelium-free fungal cultures, and plant synthesis is complicated by the detection of organic compounds involved in the reduction and stabilization of gold nanoparticles [38]. In addition, the toxicity of the target nanoparticles requires careful in vitro and in vivo testing, especially for use as human drugs. Nevertheless, all these barriers are surmountable, and the following prospects in gold nanoparticles usage are possible.

High efficacy against pathogenic microorganisms has been confirmed by most studies. Obtaining drugs based on AuNPs having antibacterial and antifungal effects is extremely promising in light of the high pathogen resistance to antibiotics. Alternative methods would help to pass this problem. Moreover, biogenic capping agents with antimicrobial activity can enhance the desired effect. Recent work on antiviral activity proves that the adsorption of AuNP due to van der Waals forces on virion spikes can disrupt the attachment of the virus to cellular receptors and prevent penetration into the cell [287]. Based on the understanding of such mechanisms, new antiviral drugs can be created. Although the antiviral activity of biosynthesized gold nanoparticles has not been studied as intensively as silver nanoparticles [288,289,290], this approach is also interesting and needs further development.

Due to the antioxidant properties of AuNPs, diseases highly dependent on oxidative stress caused by ROS (inflammatory, atherosclerosis, aging, cancer) can be prevented. On the other hand, a huge number of publications are devoted to the anticancer activity of gold nanoparticles. High biocompatibility and biodegradability have increased the utility of biosynthesized gold nanoparticles in cancer therapy [85]. The potential use of nanoparticles encrusted with antitumor compounds (for example, capping agents from medicinal herb extracts or functionalized by chemical medicines) will increase the effect of drugs. The effect on many cell lines gives hope for obtaining drugs with low toxicity and high anticancer efficiency. Research also should start to focus on in vivo studies.

The diabetic problem in the world is very acute, and the continued research on the antidiabetic properties of gold nanoparticles is very relevant. An extremely interesting direction is the larvicidal activity of AuNPs. The high efficiency of AuNPs in killing the larvae gives a possibility to use them as safe drugs instead of expensive and polluting insecticides [291]. Most probably, new potentially useful properties of gold nanoparticles will be discovered soon. For example, AuNPs from *Crataegus oxyacantha* extract have potent urease enzyme inhibitors activities [292].

High affinity on the surface of AuNPs gives wide application possibilities in drug-delivery systems. Gold nanoparticles can be incorporated into biosensors to increase their stability, sensitivity, and selectivity. AuNPs can be used as detectors of pathogenic microorganisms. Several recent studies are devoted to developing various advanced schemes for virus detection with the help of AuNP [282]. AuNP-based nucleic acid assays for the detection of severe acute respiratory syndrome (SARS) [293], AuNP-based scanometric and surface-enhanced Raman scattering (SERS) for the Ebola virus detection [294], AuNP-based assays for hepatitis C virus (HCV) detection [295] is discovered.

The approaches described above are the most famous in the application of gold nanoparticles. Nevertheless, the range of applications is regularly expanded. AuNPs have an anticoagulant effect in blood plasma that will contribute to medicine in controlling thrombotic disorders [296]. According to inflammatory and antioxidant characteristics, AuNPs can be used to treat neurodegenerative diseases: chronic brain diseases associated with tauopathy, neuroinflammation, and oxidative stress in the cortex and hippocampus [297,298]. AuNPs suppress macrophage and microglial activation in the brain and reduce TNF-α levels in the hippocampus [298,299]. In neurodegenerative disease, AuNPs were shown to suppress the pro-inflammatory responses in a microglial cell line, which is beneficial for the central nervous system repair and regeneration [299]. AuNPs can be a therapeutic drugs carrier, which are more effective as anti-inflammatories than AuNPs or drugs alone [300]. Gold nanoparticles may be useful in the treatment of Alzheimer’s disease, as they can suppress amyloidosis through the effect on the Aβ (amyloid-β) process of aggregation and fibrillation [301]. Further research in this area may become a novel strategy in the creation of anti-amyloid drugs. The study of anthelmintic activity is very promising. The effectiveness of gold nanoparticles from phytopathogenic fungus *Nigrospora oryzae* was demonstrated in a plausible anthelmintic role as vermifugal agents against a model cestode *Raillietina* sp., an intestinal parasite of domestic fowl [302]. Antileishmanial and antiplasmodial activities, which are extremely important in the light of the fight against tropical infections, were demonstrated [303,304]. Very interesting data were obtained about analgesic and antispasmolytic activity [305]. The involvement of μ-opioid receptors mediated by AuNPs-from *Euphorbia wallichii*, resulting in the generation of analgesic response through the central system, was shown. In addition, the effect may be associated with capping agents—alkaloids, flavonoids, and saponins, which have anti-inflammatory and analgesic properties [305]. Owing to simple surface functionalization and excellent biocompatibility, AuNPs modified with proteins, peptides, and DNA are used in vaccines [306].

Summing up the above, the variety of AuNPs’ properties can make them indispensable assistants in the fight against diseases of the most diverse origin. Studying the mechanisms of “green” AuNP’s effect on living cells can not only bring us closer to solving a considerable number of modern medical problems but also expand the application horizons of these amazing nanoparticles.

## Figures and Tables

**Figure 1 jfb-12-00070-f001:**
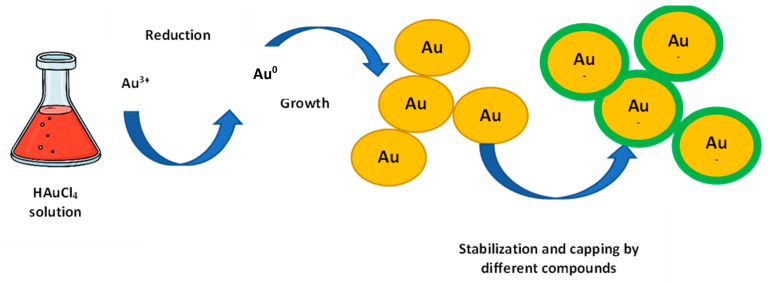
The mechanism of AuNPs biosynthesis.

**Figure 2 jfb-12-00070-f002:**
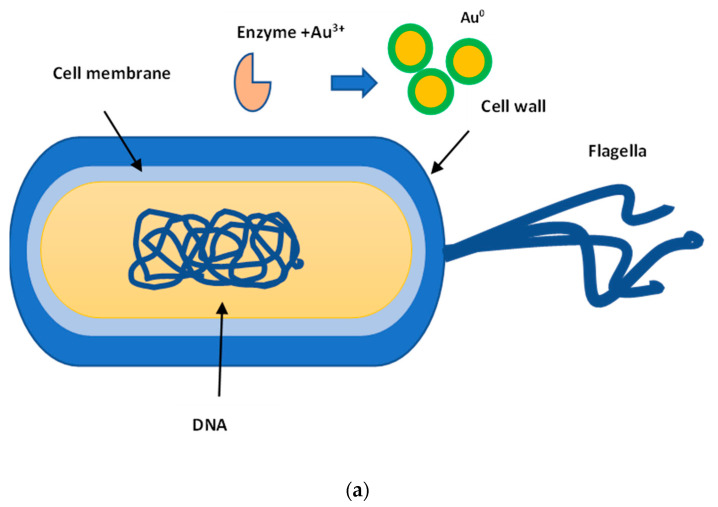
Schematic mechanism of extracellular (**a**) and intracellular (**b**) AuNPs biosynthesis.

**Figure 3 jfb-12-00070-f003:**
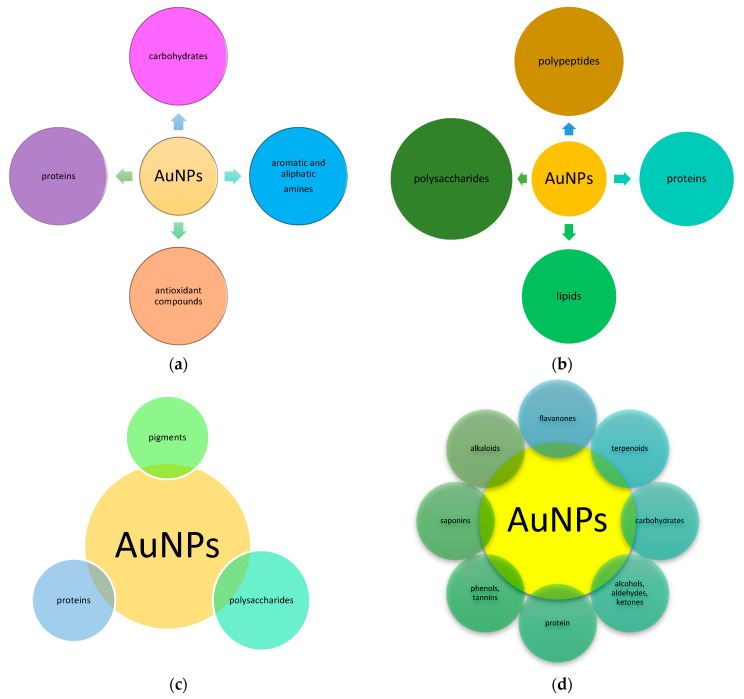
Capping agents of AuNPs: (**a**) bacterial; (**b**) fungal; (**c**) algal; (**d**) plant.

**Figure 4 jfb-12-00070-f004:**
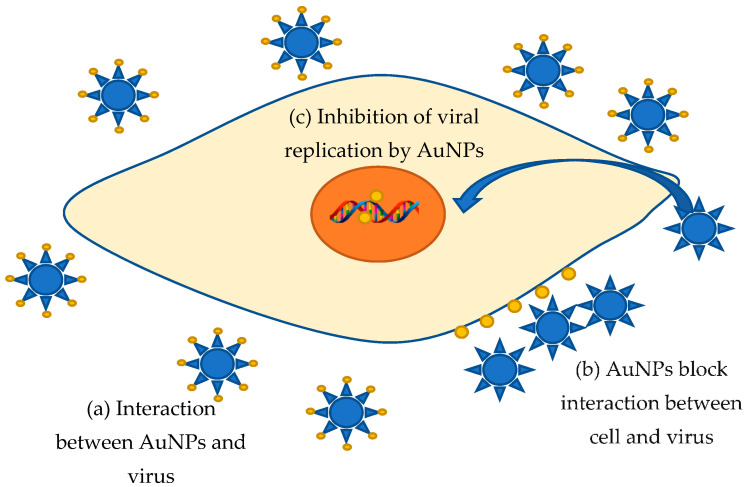
Proposal mechanism of AuNPs antiviral activity.

**Table 1 jfb-12-00070-t001:** AuNPs synthesized by bacteria.

Microorganism	Shape	Size, nm	References
*Stenotrophomonas maltophilia*	spherical	40	[41]
*Rhodopseudomonas capsulata*	spherical	10–20	[42]
*Pseudomonas putida and Pseudomonas fluorescence*	spherical	10–50	[43]
*Deinococcus radiodurans*	spherical, pseudo-spherical, truncated triangular and irregular	~43.75	[49]
*Bacillus cereus*	spherical, hexagonal, and octagonal with irregular contours	40–50	[92]
*Marinobactor pelagius*	varied	~2–6	[93]

**Table 2 jfb-12-00070-t002:** AuNPs synthesized by fungi.

Microorganism	Shape	Size, nm	References
*Pycnoporus sanguineus*	spherical, pseudo-spherical, triangular, truncated triangular, pentagonal, and hexagonal	several to several hundred	[94]
*Magnusiomyces ingens*	spherical, triangular, hexagonal	10–80	[95]
*Thermoascus thermophilus*	different	~10	[96]
*Trichoderma hamatum*	spherical, pentagonal and hexagonal	5–30	[97]
*Aspergillus foetidus*	spherical	30–50	[98]
*Rhizopus oryzae*	spherical	5–65	[99]

**Table 3 jfb-12-00070-t003:** AuNPs synthesized by algae.

Organism	Shape	Size, nm	References
*Stoechospermum marginatum*	spherical, hexagonal and triangle	18–93	[98]
*Laminaria japonica*	spherical	15–20	[99]
*Ulva fasciata*	spherical	~10	[100]
*Chlorella vulgaris*	spherical	2–10	[101]
*Prasiola crispa* (green algae)	spherical	5–25	[102]
*Galaxaura elongata*	rod, triangular, truncated triangular and hexagonal	3–77	[103]
*Turbinaria conoides* (brown algae)	small spherical, triangle and pseudo-spherical	6–10	[105]
*Sargassum polycystum* (brown algae)	spherical	68–240	[106]

**Table 4 jfb-12-00070-t004:** AuNPs synthesized by plants.

Plant	Shape	Size, nm	References
*Gymnocladus assamicus* (pod extract)	hexagonal, pentagonal, and triangular	4.5–22.5	[71]
Areca catechu nut	spherical	~13.7	[72]
*Croton Caudatus Geisel* leaf extract	spherical	20–50	[73]
*Petroselinum crispum* (leaf extract)	spherical, semi-rod, flower shaped	17–50	[74]
*Salix alba L*. leaves extract	-	50–80	[75]
*Sesbania grandiflora* leaf extract	spherical	7–34	[78]
*Mimosa tenuiflora* bark extract	multiple	20–200	[79]
*Terminalia chebula* seed powder	pentagonal, triangular, spherical	6–60	[80]
*Jasminum auriculatum* leaf extract	spherical	8–37	[81]
*Solanum nigrum* leaf extract	spherical	~50	[82]
*Citrullus lanatus* rind extract	spherical	20–140	[87]
Mango peel extract	spherical	6–18	[104]
*Mentha piperita* leaf extract	hexagonal	~78	[105]
*Coleus aromaticus* leaf extract	spherical, rod, and triangular	~80	[106]
*Anogeissus latifolia* leaf extract	spherical	50–60	[107]
*Papaver somniferum* seed pulp extract	spherical	~77	[108]
*Aloysia triphylla* leaf extract	spherical	40–60	[109]
*Trigonella foenum*-graecum seed extract	-	15–20	[110]
*Punica Granatum* fruit extract	triangular and spherical	5–20	[111]
*Eucommia ulmoides* bark aqueous extract	spherical	~18.2	[112]
*Capsicum annuum var. grossum* pulp extract	triangle, hexagonal, and quasi-spherical	6–37	[113]
*Plumeria alba* flower extract	spherical	15–28	[114]
*Platycodon grandiflorum* leaf extract	spherical	~15	[115]
Siberian ginseng	spherical	~200	[116]
*Marsdenia tenacissima*	spherical	~50	[117]
*Peganum harmala* seed extract	spherical	43–52	[118]
*Garcinia mangostana* fruit peel extract	spherical	~32	[119]

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
