# Peer review of "Gold Nanoparticles: Biosynthesis and Potential of Biomedical Application"

_jfb, 2021, doi:10.3390/jfb12040070_

Round 1
Reviewer 1 Report
The manuscript sets out to review the biomedical applications of Gold Nanoparticles. There are many similar reviews available, but what potentially sets this review apart is the focus on ""green" biosynthesis of gold nanoparticles by bacteria, fungi, algae, and plants". This focus is mentioned in the abstract, but is not evident from the title, which should be revised to reflect this. Some other aspects which should be considered before publication:
(i) The abstract should be extended to provide a reader, who only has access to the abstract, with a more detailed outline of what is contained in the paper.
(ii) AuNP is an acronym, not a Keyword. The paper also covers many other topics which should be included as Keywords (antimicrobial etc.)
(iii) The language of the manuscript shoudl be thoroughly revised, ideally by a native speaker.
(iv) Introduction: "The metal range used to create these amazing objects is very large: there are publications about silver, gold, platinum, nickel, manganese, titanium, zinc nanoparticles, and others." Nanoparticles are not always metallic. References should be provided for "there are pubications.."
(v) "In spite of the fact, silver nanoparticles occupy the lion's share of this topic research, other nanoparticles also seem to be interesting objects. Gold nanoparticles are undoubtedly the second most popular nanoparticles." - please provide references
(vi) "In the Middle Ages, with the alchemy prosperity, gold was attributed magical and healing properties: the ability to cure heart diseases, infectious diseases, cancers and generally have a beneficial organism effect." - please provide references
(vii) "Fourier-transform infrared spectroscopy (FTIR) give a possibility to directly visualize gold nanoparticles" - how does FTIR visualise gold nanoparticles? Please provide references
(viii) "Possessing unique physicochemical properties, including surface plasmon resonance (SPR)" - SPR is not unique to gold nanoparticles, and indeed not unique to nanoparticles.
(ix) 2. AuNPs Biosynthesis - it would be better to first desceibe the general synthesis of AuNPs, and then explain how this synthesis is achieved by biological systems, in a "green" fashion.
(x) 2. AuNPs Biosynthesis - references are missing in multiple places.
(xi) "The general scheme assumes the following: the biological fluid is added drop by drop" - what biofluid? Blood? Sweat? Tears?
(xii) "The sizes of gold nanoparticles are significantly smaller than sizes of biological objects," - really? viruses? mitochoncria? exosomes? The authors should be more precise.
(xiii) "They have unique properties - catalytic, ferromagnetic, optical properties." - are these really unique?
(xiv) "These characteristics are based on the phe-183 nomenon of surface plasmon resonance (SPR)." - explain this further and/or provide references.
(xv) "The smaller size of the nanoparticles leads to an increase in the surface area." - this is not true.
(xvi) "... action with various cellular organelles, including DNA" - DNA is not an organelle
(xvii) "...3D nanoparticles..." - all nanoparticles are 3D.
(xviii) Viruses are not considered micro-organisms.
(xix) "It is assumed that AuNPs can bind to a viral particle, blocking the connection with cellular receptors or viral receptors that inhibit viral cycle onset (fig. 2)." - what evidence is there for this? Or who assumes?
(xx) The Conclusions section should explore the future perspectives. What is required to bring the use of AuNPs into more everyday and/or clinical use for the range of applications identified.
(v)
Author Response
Dear colleague,
Thank you very much for your attentive attitude to my publication and valuable comments. All necessary revisions are highlighted in yellow.
(i) The abstract was improved and more detailed (highlighted in yellow). The title of the article was also amended.
(ii) Key words were added (highlighted in yellow).
(iii) The language of the whole manuscript was revised.
(iv-vi) The introduction section was completely reworked and more references were added (highlighted in yellow).
(vii) Information and references about FTIR analysis were provided (highlighted in yellow).
(viii) This phrase was deleted and replaced with a more appropriate.
(ix) According to Your revision I reorganized this section: at first, comparative analysis with chemical and physical methods, as well as the general mechanism were discussed, at second, biological synthesis was displayed (highlighted in yellow).
(x) 2. Different references in section “AuNPs Biosynthesis” were added (highlighted in yellow).
(xi- xiii) These phrases were deleted and replaced with a more appropriate in section “The AuNPs morphology (shape and size)”, references were added.
(xv) This phrase was deleted.
(xvi) This phrase was corrected in section “Mechanism of AuNPs action on cells” (highlighted in yellow).
(xvii) This phrase was deleted.
(xviii) This sentence was rephrased in section “Antiviral activity”.
(xix) The proposed mechanism of AuNPs antiviral activity was presented according to the references presented in the section “Antiviral activity” (highlighted in yellow).
(xx) The Conclusions section was completely rewritten and the future perspectives were presented (highlighted in yellow).
Reviewer 2 Report
This review paper deals with Gold nanoparticles: possibilities and prospects of biomedical application. Its review is very interesting work in biomedical for potential readers. However, there is not clear something for publication and I recommend the author should improve the manuscript.
1. Author shows well-summarized technologies related to gold particles in the table to easily understand. However, the manuscript gives only two images for potential readers compared to review papers. I strongly recommend author should add various images for example.
2. This manuscript consists of 10 themes. You should reorganize contents under similar topics. For example, 1. Introduction--> 2. Properties of gold nanoparticles: 2.1. The AuNPs morphology (shape and size), 2.2 10. Toxity, 2.3 Capping and stabilizing agents --> 3. Activity …. --> 4. ....--> 5. --> 6. Conclusions
3. In 4. Capping and stabilizing agents, can you explain more detailed mechanism capping process?
4. Gold particles have been used for biomedical applications but you deal with some applications. I recommend you would introduce various biomedical applications such as drug delivery systems (DDS), hyperthermia, and bio-sensor, etc.
5. In conclusion, this part should show the future and development of gold particles.
Author Response
Dear colleague,
Thank you very much for your attentive attitude to my publication and valuable comments. All necessary revisions are highlighted in green, blue and yellow.
- Two images: The mechanism of AuNPs biosynthesis and Schematic mechanism of extracellular and intracellular AuNPs biosynthesis were added (highlighted in green and blue also for other reviewers).
- The sections were reorganized based on your opinion and the opinion of other reviewers (highlighted in green).
- Unfortunately, there is not much information about the capping mechanism, it is mainly about the involved molecules. But some more information was added (highlighted in green).
- Information about photothermal therapy, drug delivery, bio-sensing and detection was added (highlighted in green).
- The Conclusions section was completely rewritten and the future perspectives were presented.
Reviewer 3 Report
- Abstract is too simple, I suggest author should improve the abstract and introduce more information about background and topic of “green” biosynthesis.
- In the “introduction part”, too many meaningless information was introduced. I suggested that author should focus on history of gold nanoparticles development. Moreover, author should cite references to support your points.
- Author should be well check the format of this manuscript, because too many tiny mistakes can be observed, for example line 86.
- I suggested that author should divide “biosynthesis section” into two different section: for example, like mechanism, and applications.
- Compared to chemical synthesis, what is the advantage of AuNP biosynthesis? I did not get any point to highlight this problem.
- Can author provide a schematic illustration about intracellular and extracellular biosynthesis? Because scheme can provide more clear presence about these processes.
- Line 215, bacterial AuNPs, I think author may want to point about protein corona?
- Line 264, too many papers reported toxicity of AuNPs, for example Scientific reports, 2019, 9(1): 1-19.; Environmental Toxicology and Chemistry, 2020, 39(12): 2450-2461.
- Section 5, antimicrobial activity of AuNPs, I thinks that this particle descript toxicity about AuNPs not antimicrobial activity. Author should well re-organize this section.
- AuNPs cannot generate ROS, and major resource of ROS is from capping molecules, thermal effects and Au ions.
- Antioxidant will inhibit the anticancer activity of NPs. Authors should well provide clear view point to identify what can lead to antioxidant and what can lead to anticancer.
- In every section, author should concisely descript backgroup.
Author Response
Dear colleague,
Thank you very much for your attentive attitude to my publication and valuable comments. All necessary revisions are highlighted in yellow and blue.
- The abstract was improved and more detailed (highlighted in yellow according to other reviewers). The title of the article was also amended.
- The introduction section was completely reworked and more references were added (highlighted in yellow also for other reviewers).
- I tried to carefully check the text to correct errors.
- Biosynthesis section was reorganized: at first, comparative analysis with chemical and physical methods, as well as the general mechanism were discussed, at second, biological synthesis was displayed (highlighted in yellow also for other reviewers).
- The advantages of AuNP biosynthesis were presented in Biosynthesis and Conclusion sections (highlighted in yellow also for other reviewers).
- Schematic mechanism of extracellular(a) and intracellular (b) AuNPs biosynthesis was presented (highlighted in blue).
- The information about protein corona was added (highlighted in blue).
- The proposed publication (Scientific reports, 2019, 9(1): 1-19) contains a lot of useful information, unfortunately I did not find the second article in the open access. Data was added to the section “Mechanism of AuNPs action on cells”.
- According to Your revision, I reorganized section “antimicrobial activity”. I think it would be better to say about Mechanism of AuNPs action on cells. This section is divided to two sub-sections: toxity for bacterial cells and toxity for human cells (highlighted in blue). A separate section was devoted to antimicrobial activity (antibacterial and antifungal) ((highlighted in blue).
- I agree, that major resource of ROS is from capping molecules, thermal effects and Au ions. The role of ROS in AuNPs toxity was described in section “Anticancer activity” (highlighted in blue).
- More information about antioxidant activity was added to section “Antioxidant activity”. Also, antioxidant activity was discussed early than anticancer activity in order to present the material more clearly (highlighted in blue).
- The Conclusions section was completely rewritten and the future perspectives were presented (highlighted in yellow also for other reviewers).
Round 2
Reviewer 1 Report
The author has satisfactorily addressed most of the issues raised during the initial review. However, the language still requires substantial improvement.
Author Response
Dear colleague,
Thank you very much for your attentive attitude to my paper and useful comments that contributed to improving its quality. The grammar and style of the text were checked by person who know English very well, and some adjustments were made. In this regard, we would like to note that the other two reviewers did not make any comments about the English quality. If our esteemed reviewer still has doubts about the English quality, then we think to leave the final decision on this matter to the editors of journal.
Reviewer 2 Report
The authors have been well revised your manuscript. However, I strongly recommend you would add other research works.
In your manuscript, you have been shown good images to easily understand technologies for gold nanoparticles. But I can't find some results that is including experimental values and results.
Conventional review papers should have some results of enhanced performances, new shapes, or progress compared to other previously published researches.
Author Response
Dear colleague,
Thank you very much for your attentive attitude to my paper and useful comments that contributed to improving its quality. Unfortunately, it remains unclear what our esteemed reviewer means by "some results, including experimental values and results", because there is no indication of this in repeated review. If we are kindly provided with information about such publications that were not listed in the review, we will be happy to cite them in the article and add to the references. However, some information not previously provided has been added to the section Conclusions. All necessary revisions are highlighted in green.
Reviewer 3 Report
Author have well-revised this review on the basis of previous suggestions. I suggested to be accepted for further publishing in this journal.
Author Response
Dear colleague,
Thank you very much for your attentive attitude to my paper and useful comments that contributed to improving its quality.